# Effect of Electrode Spacing on the Detection of Coating Defects in Buried Pipelines Using Direct Current Voltage Gradient Method

Seung-Heon Choi [1], Young-Ran Yoo [2],* and Young-Sik Kim [1,2],*

[1] Department of Materials Science and Engineering, Andong National University, 1375 Gyeongdong-ro, Andong 36729, Gyeongbuk, Republic of Korea; csh140541@pyunji.andong.ac.kr

[2] Materials Research Centre for Energy and Clean Technology, Andong National University, 1375 Gyeongdong-ro, Andong 36729, Gyeongbuk, Republic of Korea

* Correspondence: yryoo@anu.ac.kr (Y.-R.Y.); yikim@anu.ac.kr (Y.-S.K.); Tel.: +82-54-820-7897 (Y.-R.Y.); +82-54-820-5504 (Y.-S.K.)

**Abstract:** Buried piping is subject to soil corrosion, which can be prevented by combining coatings and cathodic protection to maximize corrosion control. However, even with both methods, coatings are subject to damage from external factors and various causes. Buried piping may expose the metal and alter the current flow, which in turn causes corrosion. Therefore, this study analyzed the effect of detection electrode spacing on the direct current voltage gradient (DCVG) magnitude formed for coated pipelines buried in the soil. The DCVG was measured using a real-time coating defect detection system. FEM model simulations were carried out, and then the result was compared to the measured DCVG magnitude. When the spacing of the detection electrodes increased, the detected signal and signal location changed. The detection reliability increased as the noise signal is eliminated at the optimum detection electrode spacing. However, the detection reliability decreased at higher selection electrode spacing as the noise signal and detected signals together were eliminated. The location of the detected signal shifted as the spacing of the detection electrodes increased due to the change in the detection reference point and signal magnitude.

**Keywords:** buried pipe; DCVG; real-time coating defect detection system; FEM simulation; electrode spacing





## 1. Introduction

Buried pipelines are pipes that are installed in the soil and primarily used to transport oil, gas, and water [1,2]. Burying pipes underground takes up less space than laying them above ground, does not obstruct the view, and is less prone to leaks as it is not affected by external factors like pipes laid above ground. Nevertheless, as these pipes age, corrosion can occur. Corrosion can be caused by soil conditions, electrochemical reactions, microbial activity, and electrical anomalies, which can shorten the lifespan of buried pipes and ultimately lead to hazards such as leaks [3,4]. When buried pipes are damaged, replacing or repairing them can be time-consuming and expensive, as the work is performed underground [5]. Methods to prevent corrosion in pipes include environmentally friendly metal selection, coating, chemical treatment, and electrical methods [6,7].

Generally, the pipe's outer and inner surfaces are coated and cathodically protected to prevent corrosion. However, in the case of pipes buried in the soil, coating damage can occur due for various reasons. The outer coating can be damaged due to poor construction, environmental changes in the soil, and accidents caused by the construction of other facilities [8].

When the metal surface of a metal pipe is exposed to a corrosive environment due to damage to the coatings, an electrochemical reaction occurs between the soil as an electrolyte



and the metal, resulting in corrosion [9]. To solve this problem, corrosion is prevented by supplying current to the pipe through a cathodic method on the outer surface of the coated pipe. The cathodic protection criteria must be maintained at a minimum of −850 mV (CSE) with a copper sulfate electrode in an IR-free (off-potential) state [10,11]. Nevertheless, localized external corrosion (pitting, crevices, intergranular, cracks, etc.) can occur on buried pipes due to various causes (deterioration, delamination, blistering, improper application, etc.) [12,13]. In order to prevent this risk, the used methods prioritize the location and repair of coating damage [14,15].

The indirect assessment of external corrosion of pipes is mainly based on the close interval potential survey (CIPS, i.e., one-electrode method) [16], which determines the condition of the cathodic protection, and the DCVG survey (i.e., two-electrode method) [17] to determine the location of the coating damage. Other methods include Area potential and earth current (APEC) [18], the Pearson survey [19], and alternating current voltage gradient (ACVG) [20].

Pipes are usually buried in the soil at a certain depth with a relatively simple structure [15]. However, pipelines in nuclear power plants and industrial complexes are buried in a wide area with multiple layers and cross-structure conditions at various burial depths [21]. Usually, in indirect evaluation, the defect detection voltage is often changed according to the depth of the buried pipe. To obtain a high defect detection signal, a high voltage is usually applied to the pipe [22,23]. On the other hand, in the case of the CIPS method, which detects coating defects with an electrode, the maximum measurement interval is less than 3.5 times the buried depth of the pipe [24]. In the DCVG method of detection, coating defects with two electrodes, the interval is less than 3 m [25]. However, there are few reports on how the spacing of detection electrodes affected detection reliability. Therefore, this study investigated the effect of electrode spacing on the reliability of coating defect detection in buried pipes using the DCVG method.

## 2. Materials and Methods

### 2.1. Setup of Test Bed

In this study, real-time coating defect inspection of a pipe buried in the soil was performed, and Figure 1 shows the test bed of the online coating defect detection system. River sand was used around the pipeline, and the soil up to the surface was used from the soil around the test bed, which is mainly composed of granite soil (resistivity: 24.5 kΩ·cm, pH: 6.1). The test bed consisted of a pipe buried at a depth of 1.8 m, a reference electrode, and an anode.

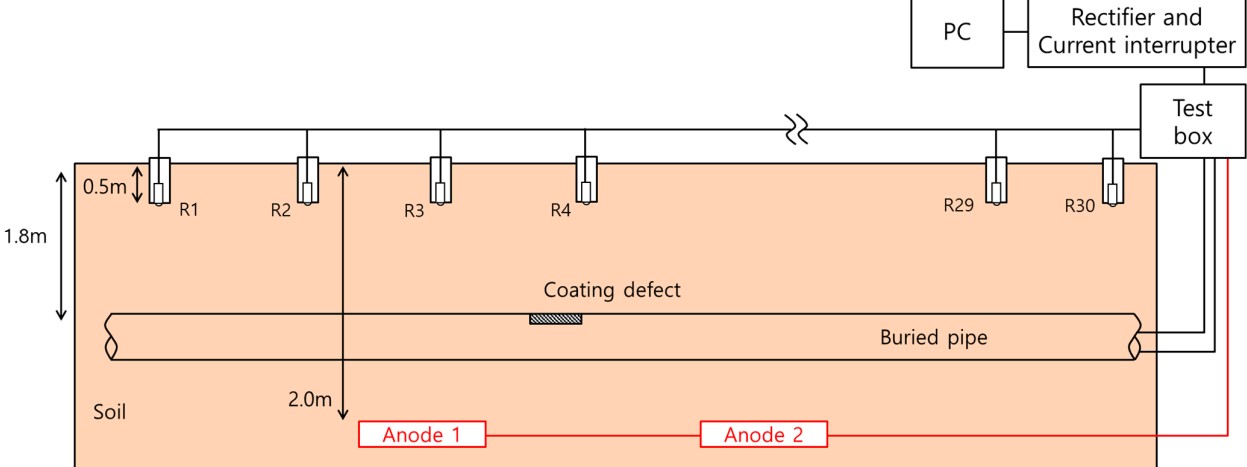

**Figure 1.** Configuration of buried pipe and cathodic protection system with real-time coating defect detection function in test bed.

The buried pipe is made of carbon steel by ASTM A106 Gr. B [26], and the length of the pipe is 30 m. It is coated with Polyken® (Seal for Life Industries, Stadskanaal, Netherlands). Polyken® coating was carried out in the following procedure: surface treatment of the pipe, primer treatment, Polyken® #930-35 lapping, and Polyken® #954-15 lapping. In this study, the coating of the pipe was intentionally removed to detection the location of the coating defects, and the size and location of the coating defects are shown in Table 1.

**Table 1.** Location and size of coating defect made intentionally on the pipe buried at a depth of 1.8 m.

| Defect Location, m | 3 | 10 | 18 | 24 |
|---|---|---|---|---|
| Defect size, cm$^2$ | 5 | 5 | 10 | 5 |

The anodes used in this study were Pt/Ti type anodes and were named 'anode 1' and 'anode 2'. These anodes were buried parallel to the pipe at a depth of 2 m below the ground surface. Using 30 detection electrodes buried at 0.5 m from the ground surface, the cathodic protection and coating defect location detection signals of the pipe were monitored in real time, and the buried detection electrodes were composed of copper–copper sulfate electrodes (Cu/CuSO$_4$, CSE). In addition, a current interrupter was utilized to measure the on- and off-potential of the pipe. The 30 detection electrodes were named 'R1' to 'R30' to measure the cathodic protection potential and coating defect detection potential, and the shielding tubes (PVC pipes) were used to protect the detection electrodes from the surrounding soil. In order to increase the accuracy of the potential measurement, insulating material was used at the entrance of the protection tube to maintain a high humidity atmosphere inside the protection tube. Figure 1 shows the configuration of the buried pipe and cathodic protection system with real-time coating defect detection function on the test bed.

### 2.2. Real-Time Coating Defect Detection System

In this study, a data logger was installed in a defect detection system to monitor the coating condition of buried pipes on a PC in real time. The DCVG (2-electrode method) measurement method was used for the coating defect detection. During the DCVG measurement process, 30 detection electrodes sequentially stored data such as on-potential, off-potential, and IR drop. The measured data were visualized in a graph on a PC.

Figure 2 shows the DCVG survey principles and defect detection by potential reversal. Figure 2a illustrates the direction and spacing of the detection electrode when detecting defects in the coating. The direction of the detection electrode is from (−) to (+), and the spacing between the detection electrodes was Δ1 m or Δ2 m or Δ3 m for each test. Figure 2b shows the DCVG results measured in Figure 2a. When the potential reversed from negative (−) to positive (+), it was considered a potential reversal signal. However, since potential reversal signals can appear regardless of the actual coating defect, to solve this problem, we defined 'detected signals' only when the location of the actual coating defect coincided with the location of the potential reversal signal. The coating defect detection reliability is calculated with Equation (1) [22].

$$\text{Detection reliability, } \% = \frac{\text{Detected signals } \times 2}{\text{Number of defect } + \text{ Number of Potential reversal}} \quad (1)$$

In the above equation, the detected signal includes all coating defect signals within an error range of ±2 m from the actual location of the coating defect [27]. The DCVG magnitude is the maximum value of the amplitude of the waveform.

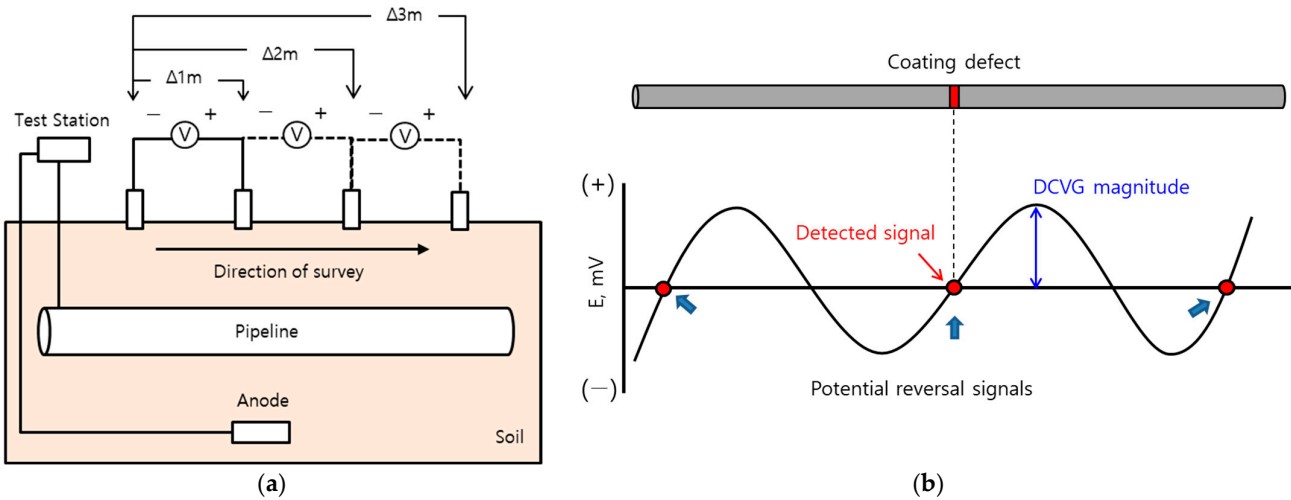

**Figure 2.** (**a**) DCVG survey principles and (**b**) defect detection by potential reversal.

### 2.3. DCVG Calculation by FEM Simulation

COMSOL Multiphysics is the program used to analyze the CIPS and DCVG signals for coating defects in buried pipes through simulation, which uses the secondary current distribution of the corrosion module. As necessary considerations for the simulation, the conductivity of the electrolyte, assumed to be activation polarization and charge transfer governed by Ohm's law at the material interface, was exposed to the electrolyte (soil). The current density and potential distribution in the electrochemical cell were calculated by the simulation, and the governing equation used in the calculation is as follows [23,28].

$$\nabla \cdot i_l = Q_l, \qquad i_l = -\sigma_l \nabla \varnothing_l$$

$$\nabla \cdot i_s = Q_s, \qquad i_s = -\sigma_s \nabla \varnothing_s$$

$$i_l = \text{Current density in liquid}$$

$$i_s = \text{Current density in solid}$$

$$Q_l = \text{Total charge in liquid}$$

$$Q_s = \text{Total charge in solid}$$

$$\varnothing_l = \text{Potential in liquid}$$

$$\varnothing_s = \text{Potential in solid}$$

$$\sigma_l = \text{Conductivity in liquid}$$

$$\sigma_s = \text{Conductivity in solid}$$

In this study, the governing equation was replaced by applying the polarization behavior in the simulated soil solution that represents the specific resistance of the test bed, and a stationary analysis was performed because it was not necessary to calculate the system change over time. The soil was assumed to be homogeneous throughout the domain.

### 2.3.1. Geometry for Modeling

Figure 3 shows the geometry of cathodic protection and defect detection for modeling. Considering the distance between the detection electrode and the pipe, the buried depth of the pipe is 1.3 m, and the anodes are located in the same place as the test bed, 1 m apart from the pipe as shown in Figure 1. The conductivity of the electrolyte, which acts as the soil in the simulation, was given a value of 0.004 S/m. The pipe was assumed to be insulated by a coating except for the intentional defects, and four defects were applied to the upper part of the pipe. The location and size of the defects are 5 cm$^2$ at 3 m, 10 m, and 24 m, and 10 cm$^2$ at 18 m, respectively.

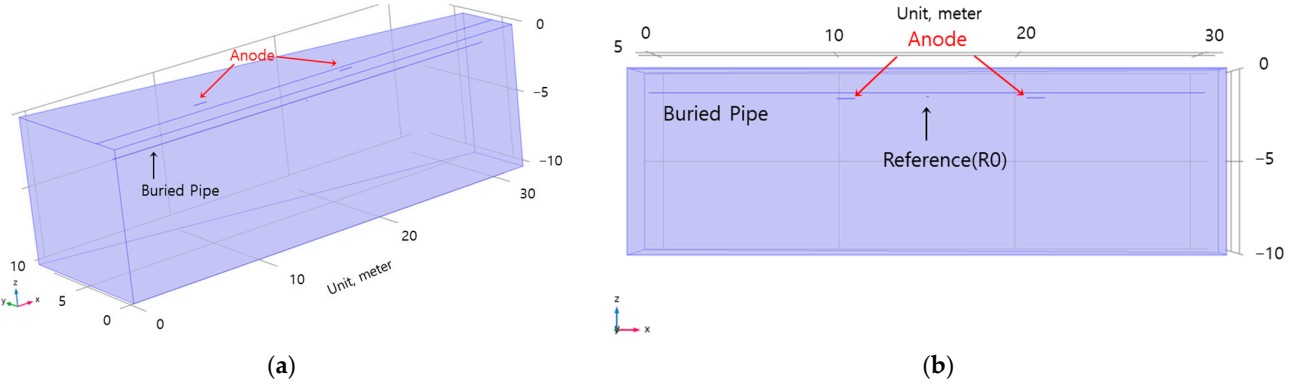

(**a**)　　　　　　　　　　　　　　　　　　　　　　　　(**b**)

**Figure 3.** The geometry of cathodic protection and defect detection for modeling; (**a**) 3D view, (**b**) Side view.

### 2.3.2. Electrochemical Parameters

Regarding the electrochemical parameters used in the modeling, the electrical conductivity measures in the soil where the pipes were buried and the corrosion potential, transmission current density, etc., were obtained through a polarization test in the simulated soil solution. Table 2 shows these parameters.

**Table 2.** Parameters for simulation runs.

| Parameter | Name | Value |
|---|---|---|
| Sigma | Electrolyte conductivity | 0.004 S/m |
| $E_{CS}$ vs. ref | Carbon steel potential vs. reference | $-1$, $-2.5$, $-3$, and $-4.5$ V(CSE) |
| $E_{eq\_CS}$ | Equilibrium potential of carbon steel | $-0.6$ V(CSE) |
| $i_{o\_CS}$ | Exchange current density, cathode | $3.09 \times 10^{-7}$ A/cm$^2$ |

## 3. Results

### 3.1. Effect of Electrode Spacing on the Detection Reliability of Coating Defect in the Pipe Buried in the Soil

Figure 4 shows the effect of electrode spacing and detection voltage on the DCVG signals measured by the real-time coating defect detection system using 30 detection electrodes placed on the ground surface for a pipe buried at a depth of 1.8 m. Figure 4a shows the data for each detection electrode interval when the detection voltage was 1.0 V. When the electrode spacing was Δ1 m, potential reversal signals appeared at 3 m, 7 m, 11 m, 14 m, 18 m, and 24 m. Among them, four signals coincided with the coating defect location, and two signals did not coincide. When the electrode spacing was Δ2 m, potential reversal signals appeared at 3 m, 11 m, 18 m, and 24 m, and four signals coincided with the coating defect location. When the electrode spacing was Δ3 m, the potential reversal signals appeared at 2 m, 18 m, and 24 m, and three signals coincided with the coating defect. When the electrode spacing increased at the detection voltage of 1.0 V, the DCVG signal increased, and the noise signal was eliminated.

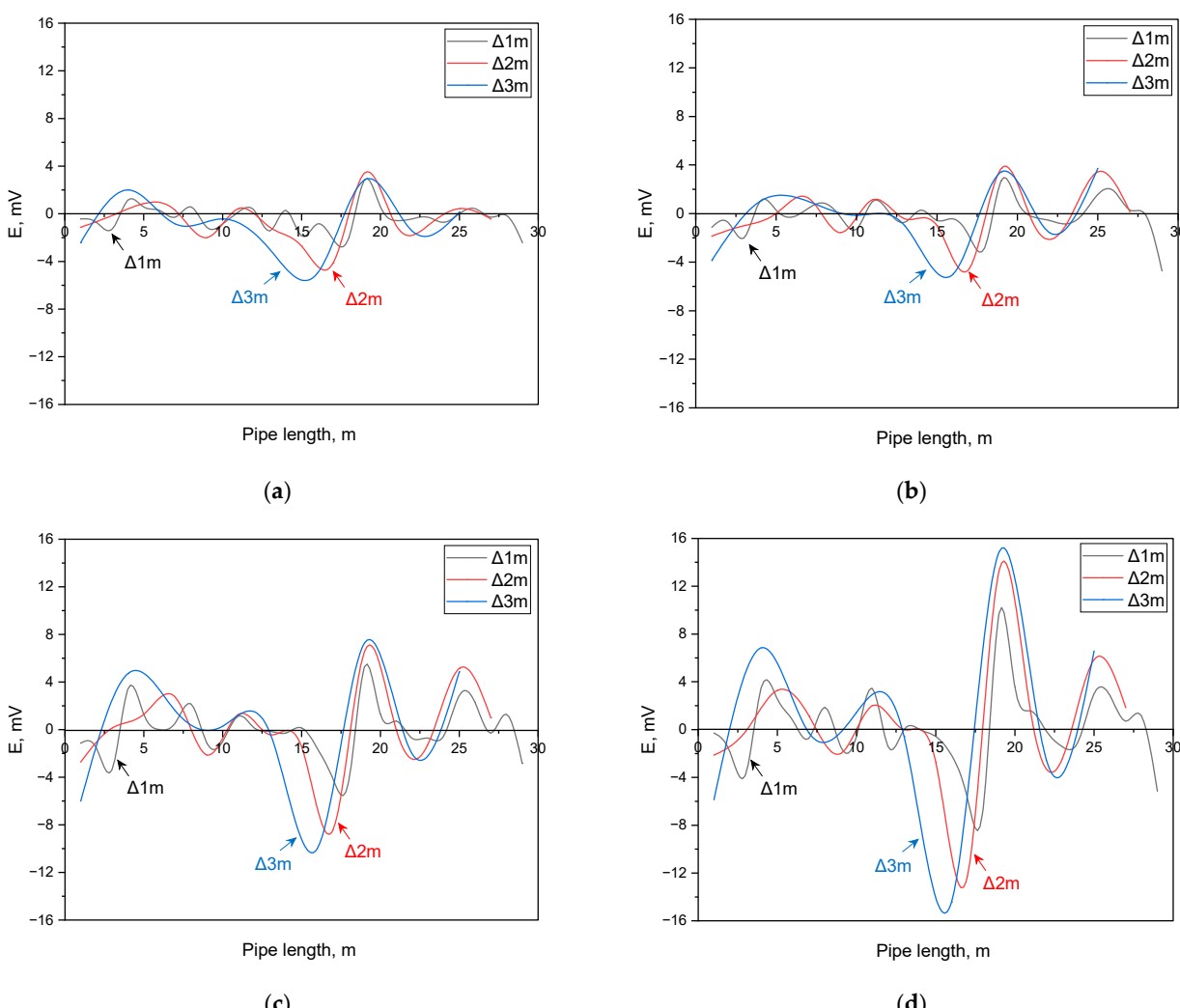

**Figure 4.** Effect of electrode spacing and detection voltage on DCVG signals measured by a real-time detection system (buried depth; 1.8 m); (**a**) 1.0 V, (**b**) 1.5 V, (**c**) 3.0 V, (**d**) 4.5 V.

Figure 4b shows a graph of the DCVG signal on the effect of electrode spacing when the detection voltage is 1.5 V. When the electrode spacing was Δ1 m, the potential reversal signals appeared at 4 m, 7 m, 14 m, 18 m, and 24 m. Among them, four signals coincided with the coating defect location. When the electrode spacing was Δ2 m, the potential reversal signals appeared at 5 m, 10 m, 18 m, and 24 m, and four signals coincided with the coating defect location. When the electrode spacing was Δ3 m, the potential reversal signals appeared at 3 m, 18 m, and 24 m, and three signals coincided with the coating defect.

Figure 4c shows the measured data measured at each detection electrode interval when the detection voltage was 3.0 V. When the electrode spacing was Δ1 m, potential reversal signals appeared at 3 m, 10 m, 14 m, 18 m, and 24 m, of which four signals coincided with the coating defect location. When the electrode spacing was Δ2 m, the potential reversal signals appeared at 3 m, 10 m, 18 m, and 23 m, and four signals coincided with the location of the coating defect. When the electrode spacing was Δ3 m, the potential reversal signals appeared at 2 m, 18 m, and 24 m, and three signals coincided with the coating defect.

Figure 4d is a graph of the DCVG signal as a result of the voltage spacing when the detection voltage is 4.5 V. When the electrode spacing is Δ1 m, the potential reversal signals appeared at 4 m, 7 m, 10 m, 18 m, and 24 m, and among them, four signals coincided with the coating defect location. When the electrode spacing was Δ2 m, the potential reversal signals appeared at 3 m, 10 m, 18 m, and 24 m, and four signals coincided with the location

of the coating defect. When the electrode spacing was Δ3 m, the potential reversal signals were 2 m, 9 m, 17 m, and 24 m, and four signals coincided with the coating defect.

As shown in Figure 4, as the detection electrode spacing increased, and the potential reversal signal decreased, while the magnitude of the defect signal and noise signal increased. At the same time, the position of the potential reversal shifted.

Figure 5 shows the effect of electrode spacing on the coating defect detection reliability calculated by the results of Figure 4 results using Equation (1). It can be confirmed that detection reliability increases regardless of detection voltage as the size of the detection electrode spacing increases. On the other hand, when the electrode spacing size is Δ3 m, the detection reliability decreases at all detection voltages except the 4.5 V detection voltage. To analyze the reasons for the increase and decrease in detection reliability, we focused on the number of potential reversals, and the results are shown in Figure 6.

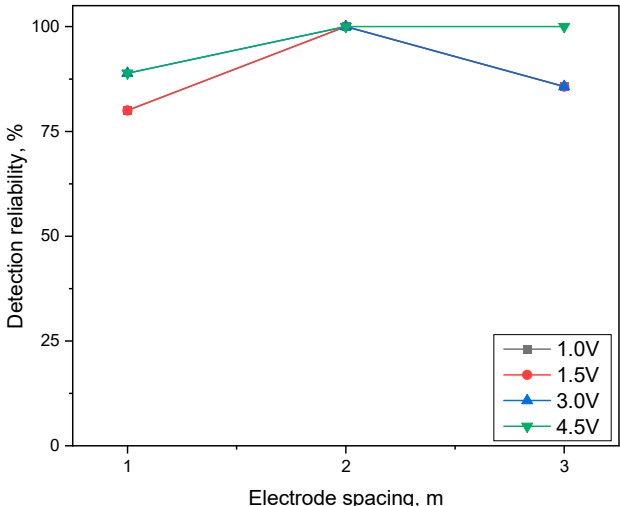

**Figure 5.** Effect of electrode spacing on the detection reliability of coating defect measured by DCVG method.

Figure 6 shows the effect of electrode spacing on potential reversal and detected signals measured by the DCVG method. Figure 6a shows the effect of electrode spacing on the number of DCVG signals when the detection voltage is 1.0 V. When the electrode spacing was Δ1 m, six potential reversal signals appeared, four of which were detection signals, and two did not coincide with the defect location. The phenomenon of detecting a potential reversal signal estimates over the number of coating defects due to interference by the stray current. On the other hand, when the electrode spacing was Δ2 m, there were four potential reversal signals and four detected signals. When the electrode spacing was Δ3 m, there were three potential reversal signals and three detected signals, but one defect was not detected. That is, when the detection voltage is as low as 1.0 V, increasing the detection electrode spacing will increase the detection reliability due to the disappearance of the noise signal. However, when the detection electrode spacing increases up to Δ3 m, the detection signal is eliminated, reducing the detection reliability even with a high detection voltage.

Figure 6b shows the effect of the electrode spacing on the number of DCVG signals when the detection voltage is 1.5 V. The results when the detection voltage was 1.5 V showed the same tendency as when the detection voltage was 1.0 V. Figure 6c shows the effect of electrode spacing on the number of DCVG signals when the detection voltage is 3.0 V. When the detection electrode interval was Δ1 m, five potential reversal signals appeared, but four were the detected signals. The detection electrode spacing of Δ2 m and Δ3 m, the potential reversal, and detected signals showed the same results as above. Figure 6d shows the effect of the electrode spacing on the number of DCVG signals when the voltage is 4.5 V. When the detection electrode interval was Δ1 m, five potential reversal signals appeared, but four were the detected signals. At the electrode spacing Δ2 m and

Δ3 m, four potential reversals and four detected signals appeared. When the detection voltage was 4.5 V, the detection reliability increased as the detection electrode spacing increased because only the noise signal disappeared.

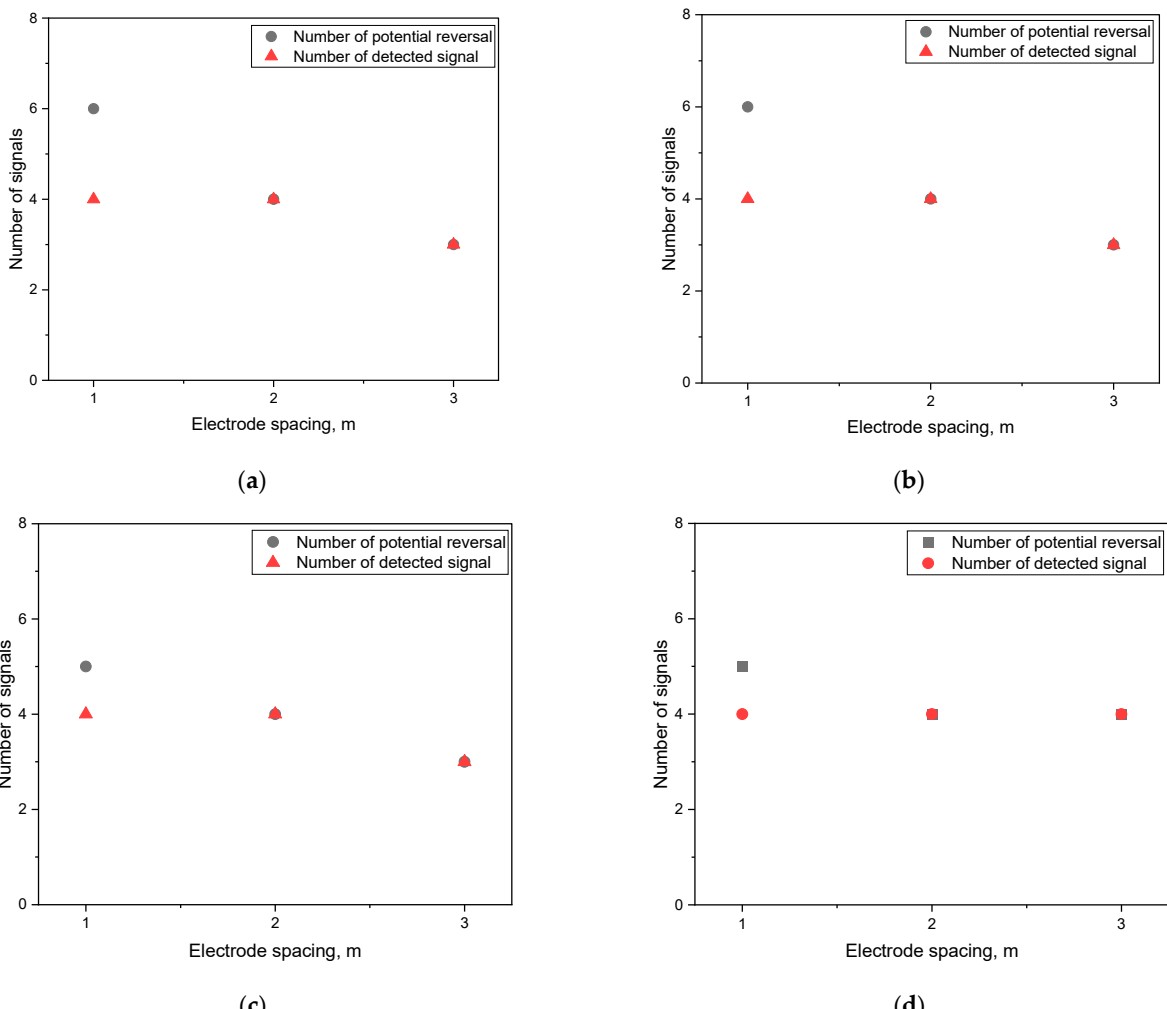

**Figure 6.** Effect of electrode spacing on the potential reversal and detected signals measured by DCVG method; (**a**) 1.0 V, (**b**) 1.5 V, (**c**) 3.0 V, (**d**) 4.5 V.

At a constant electrode spacing, the potential reversal signal and signal-to- noise magnitude increased as the detection voltage increased. However, when the electrode spacing exceeded a particular interval, the potential reversal and noise signals disappeared. On the other hand, the potential reversal position shifted as the spacing increased.

### 3.2. DCVG Calculation by FEM Simulation

To indirectly investigate the influence of detection electrode spacing and detection voltage on the detection of coating defects by the real-time detection system in buried pipes, the current distribution was calculated using the FEM method under detection conditions. Figure 7 shows the simulation results of current density and potential under the same conditions as the detection conditions in Figure 4. The simulation was performed with two anodes, and various detection voltages were applied to the buried pipe. In Figure 7a,b, it can be seen that the current from the anode directs to each coating defect of the pipe. Figure 7c,d show the pipe and line graph for the soil potential distribution on the ground surface. In this figure, the potential rises and shows a cone shape at the location of the coating defect.

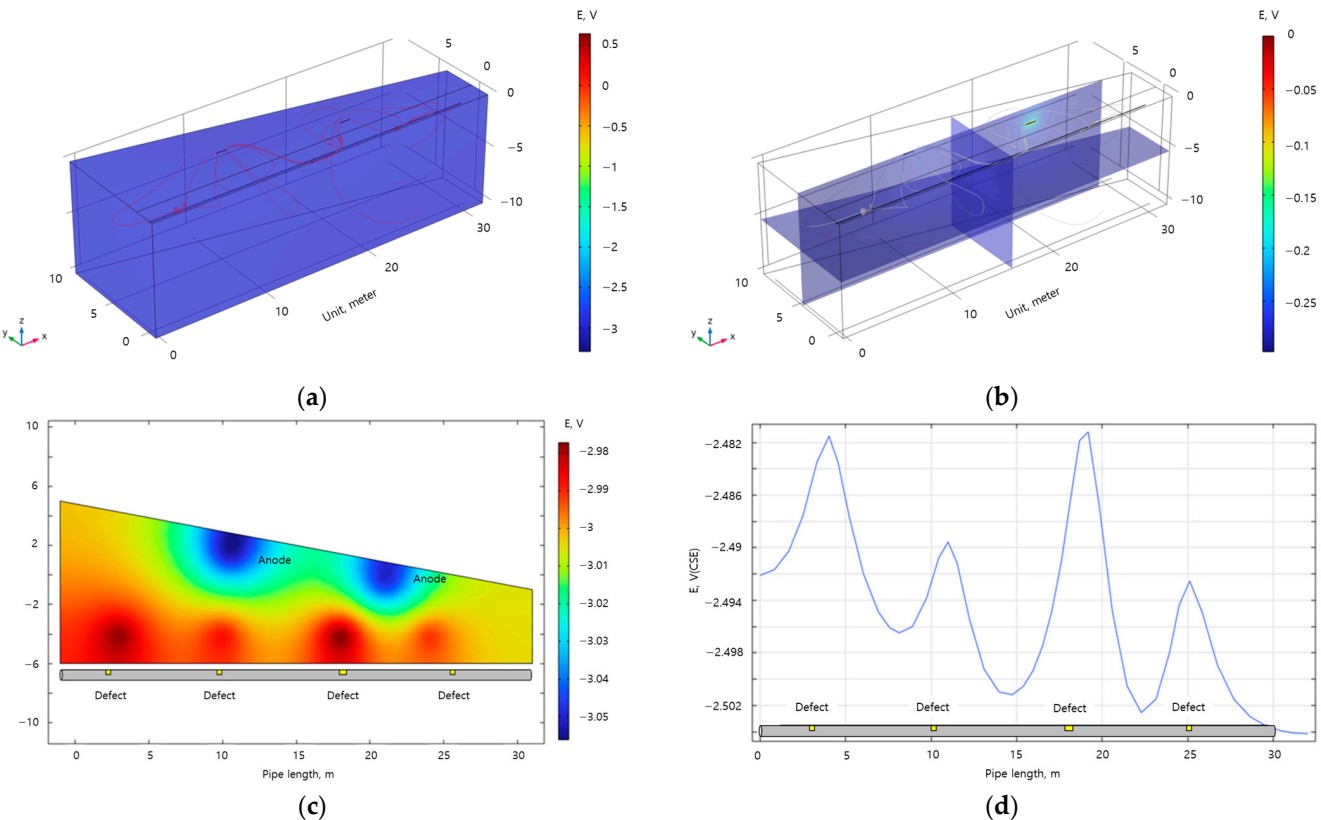

**Figure 7.** Typical simulation results for the potential and the current density on the geometry of Figure 3; (**a**) electrolyte current density, (**b**) electrolyte current density vector according to streamline, (**c**) mapping image of pipe to soil potential, (**d**) line graph of pipe to soil potential.

Figure 8 shows the DCVG data calculated by the simulation with the FEM mentioned above method and the effect of detection voltage and electrode spacing on DCVG signals measured by the simulation for a pipe buried at a depth of 1.8 m. In addition, Figure 8a shows the results when the detection voltage is 1.0 V, and the same results were obtained for all detection electrode spacings, with the number of potential reversals and the number of detected signals being four. Figure 8b–d show the results for detection voltages of 1.5 V, 3.0 V, and 4.5 V. The same results were obtained for all electrode spacings, with the number of potential reversals and detected signals being four.

The DCVG measurements calculated from the simulation all showed detection signals at the location of the coating defect. As the detection voltage and detection electrode spacing increased, the signal magnitude also tended to increase. However, the potential reversal signal location was slightly displaced from the coating defect location. Figure 4 shows that the trend was like the data results measured by the real-time defect detection system.

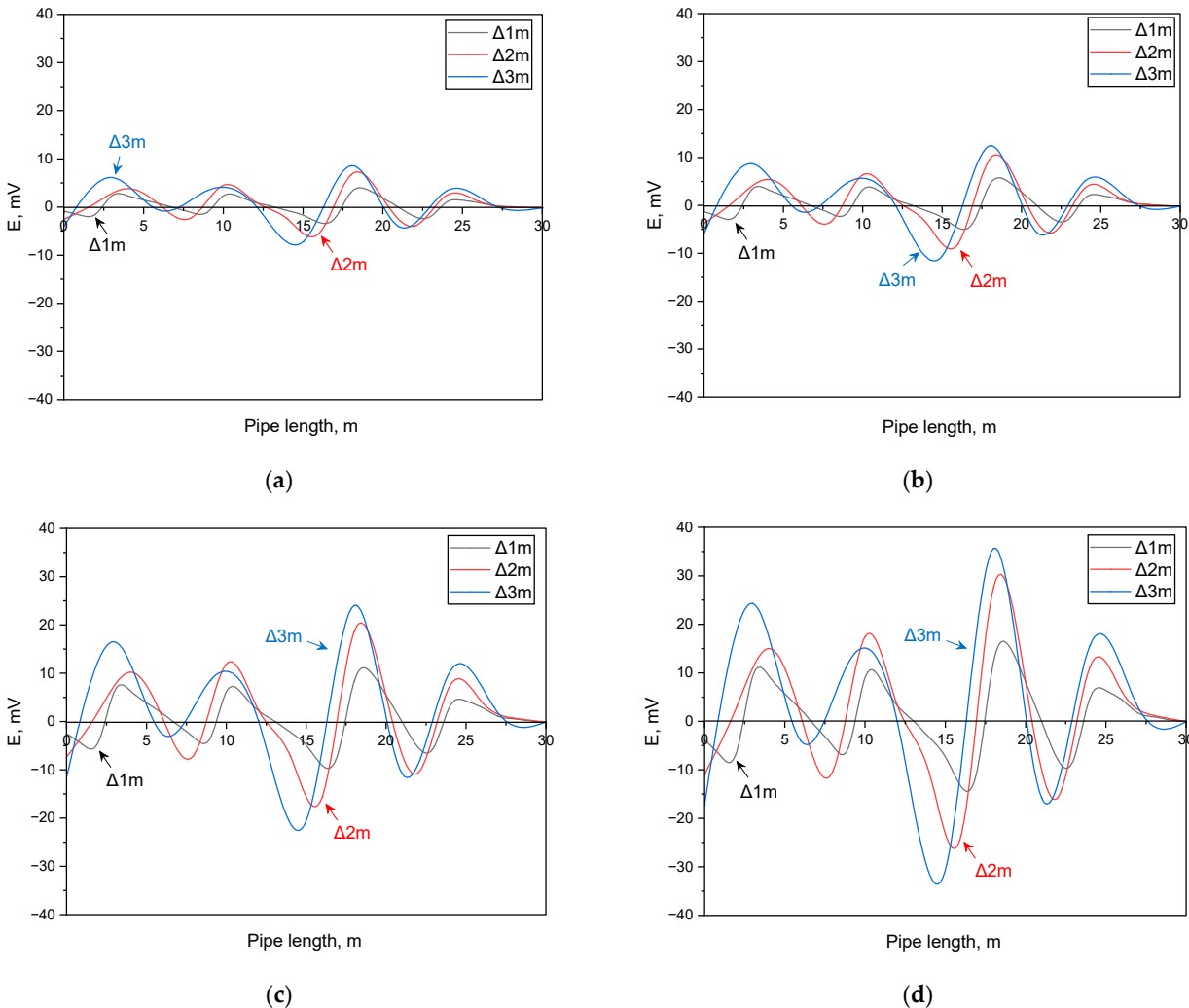

**Figure 8.** Effect of electrode spacing and detection voltage on DCVG signals measured by the simulation (buried depth; 1.8 m); (**a**) 1.0 V, (**b**) 1.5 V, (**c**) 3.0 V, (**d**) 4.5 V.

## 4. Discussion

The results of the defect signal detection on the influence of the electrode spacing and the detection voltage showed that the detection reliability tends to increase as the detection voltage and electrode spacing increase, which may result from the noise signal reduction. It was determined that the noise signal was eliminated because the magnitude of the signal increased as the spacing between the detection electrodes increased. Therefore, this study analyzed the effect of detection electrode spacing and detection voltage on DCVG detection signal magnitude, focusing on the potential reversal signal size. Figure 9 shows the results of the analysis.

Among four defects in Table 1, the defect in the 18 m analysis is to demonstrate the effect of electrode spacing on DCVG magnitude. Figure 9a shows the data obtained by the real-time detection system, and Figure 9b shows the data obtained by the FEM simulation. In Figure 9a, as the electrode spacing increases, the calculated DCVG signal magnitude increases regardless of the detection voltage, and the coefficient of determination also increases. The coefficient of determination was calculated to be 0.67 at 1.0 V, 0.57 at 1.5 V, 0.90 at 3.0 V, and 0.91 at 4.5 V detection voltage. In Figure 9b, regardless of the detection voltage, the DCVG signal increased as the electrode spacing increased, and the coefficient of determination also increased, consistent with the real-time detection system. The coefficient of determination was calculated to be 0.97 at all detection voltages.

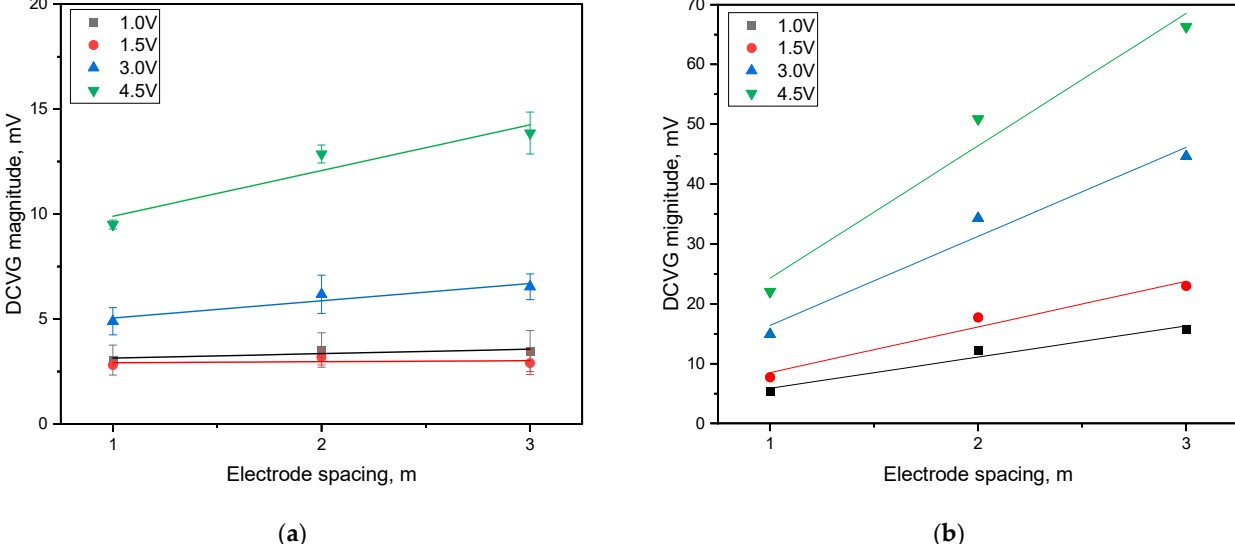

**Figure 9.** Effect of electrode spacing on DCVG magnitude obtained for the 18 m location's defect; (**a**) measured data, (**b**) simulation data.

The real-time coating defect detection system and the FEM simulation showed that the DCVG signal increased as the detection electrode spacing increased. However, the maximum value of the signal magnitude was different. The maximum signal magnitude obtained from the real-time detection system was 14 mV, while the FEM simulation showed a maximum signal magnitude of 67 mV. The soil resistivity of the testbed is 25 kΩ·cm, which was converted into conductivity (0.004 S/m) and applied in the simulation. It was calculated that the soil electrolyte is homogeneous in simulation. However, the resistivity of the real soil is not homogeneous due to the presence of particles of various sizes, such as sand and voids, inside the soil.

As the detection electrode spacing increases, the magnitude of the DCVG signal amplifies. This spacing increase also changes the detection signal's location, as shown in Figure 10. Figure 10 shows the results of analyzing the effect of detection electrode spacing on the detected signal movement using the DCVG data measured in the detection system and FEM simulations. The signal movement distance averages the movement distance of defect detection data obtained at 3 m, 10 m, 18 m, and 24 m. Figure 10a shows the movement distance of the detected signal as a function of detection electrode spacing using DCVG data from a real-time detection system. As the detection electrode spacing increased, the detection signal location was related to coating defect from the (+) direction to the (−) direction, regardless of the detection voltage.

Figure 10b shows the distance moved by the detection signal as a function of electrode spacing using the DCVG data obtained in the FEM simulation. As the electrode spacing increased, the detection signal corresponding to the defect moved in the (−) direction regardless of the detection voltage.

The data obtained from the real-time detection system shows that the detection signal is shifted from the location of the defect in the (+) direction to (−), while the simulation results show that the location of the detection signal shifts to a greater extent in the (−) direction. The cause of this phenomenon was determined to be the placement of the defect detection electrode in the simulation, which was measured directly above the defect. In contrast, the detection electrode installed on the test bed was installed differentially from the location of the defect. However, both methods shifted the defect detection signal in the (−) direction from 1 to 1.5 m as the detection electrode spacing increased. This is because the reference points of the two electrodes change as the detection interval increases, and the positions of the maximum and minimum values of the DCVG signal in the waveform shift shape.

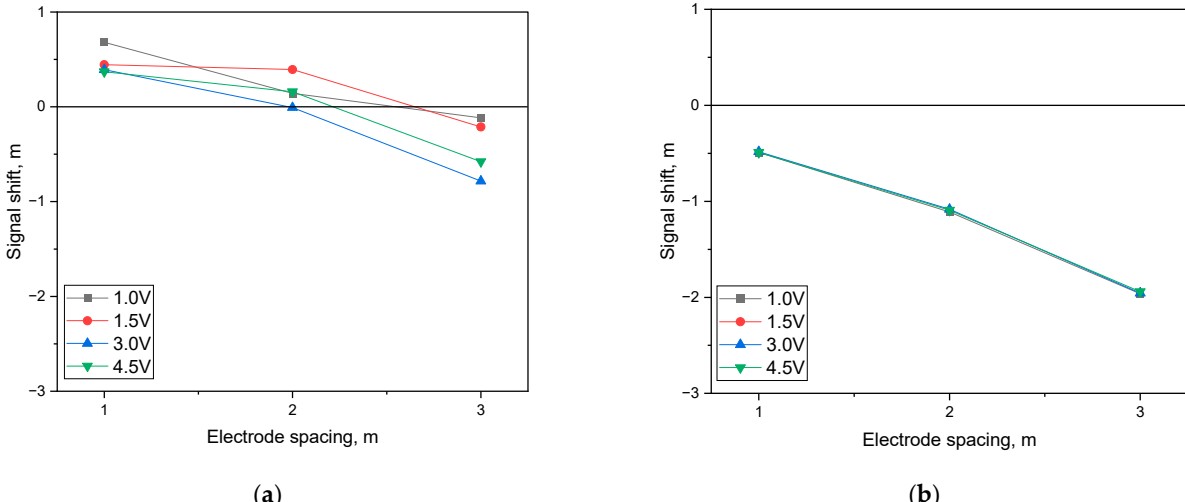

(**a**)    (**b**)

**Figure 10.** Effect of electrode spacing on the signal shift (defect location; 18 m); (**a**) real-time detection, (**b**) calculation by FEM Simulation.

As discussed above, two effects occur as the electrode spacing increases. Figure 11 shows the proposed model on the signal change of DCVG magnitude maximization and the signal shift of potential reversal.

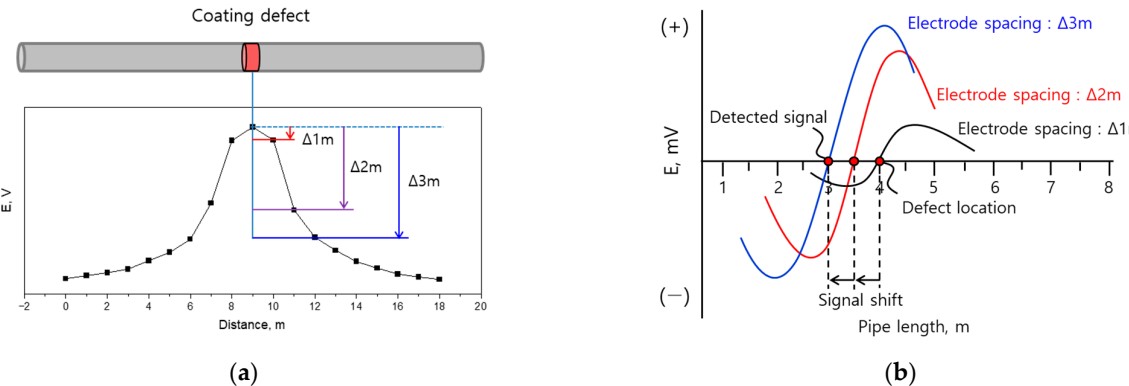

(**a**)    (**b**)

**Figure 11.** Proposed model on (**a**) the DCVG amplitude and (**b**) the signal shift of potential reversal.

First, the DCVG signal magnitude increases as the electrode spacing increases. The increase in DCVG signal magnitude by increasing the electrode spacing is due to an increased potential of defect location in the pipe. When one detection electrode is positioned slightly above the defect site, the potential difference can increase as the other electrode moves away from the defect location (Figure 11a). However, when the electrode spacing widens beyond a certain level, the signal magnitude no longer increases, and the potential difference does not change significantly.

The second is the shift of potential reversal location; When there is a defect in the pipe, the DCVG signal appears in the waveform, and as the spacing between the detection electrodes increases, the reference point of the detection electrodes changes, which causes the maximum and minimum values of the detection signal. Therefore, as show in Figure 11b, the detection signal level moves according to these changes.

## 5. Conclusions

This study analyzed potential reversal signal and detection reliability due to the change in electrode spacing by real-time detection and FEM simulation for a pipe buried 1.8 m deep in the soil. The following conclusions were drawn.

1.  The optimal spacing of the detection electrodes was found to be Δ2 m in investigating the coating defect of pipe buried in the soil. The reason is that when the electrode spacing increases, the noise signal is eliminated, and when the electrode spacing exceeds the appropriate interval, the potential reversal signal disappears with the noise signal, and the defect detection reliability also decreases.

2.  The location of the defect detection signal is shifted as the detection electrode spacing increases. This is because the reference points of the two electrodes move as the detection interval increases, and the positions of the maximum and minimum values of the DCVG signal in the waveform shift shape.

3.  As the detection electrode spacing increased, the DCVG signal magnitude increased according to both the real-time detection method and the FEM calculation method, and a similar trend of potential reversal location was obtained. However, the magnitude was more significant for the FEM calculation method. This tendency may be related to the inhomogeneity of the real soil.

This work dealt with a test bed. In the future, we plan to conduct research by applying this approach to a nuclear power plant in South Korea.

**Author Contributions:** Conceptualization, S.-H.C., methodology, Y.-R.Y. and S.-H.C., investigation, S.-H.C., data curation and analysis, Y.-R.Y., writing—original draft preparation, Y.-R.Y. and S.-H.C., writing—review and editing, Y.-R.Y. and Y.-S.K., supervision, Y.-S.K. All authors have read and agreed to the published version of the manuscript.

**Funding:** This work was supported by the Korea Institute of Energy Technology Evaluation and Planning (KETEP) grant funded by the Korea government (MOTIE) grant 20217910100010.

**Institutional Review Board Statement:** Not applicable.

**Informed Consent Statement:** Not applicable.

**Data Availability Statement:** Not applicable.

**Acknowledgments:** This work was supported by the Korea Institute of Energy Technology Evaluation and Planning (KETEP) grant funded by the Korea government (MOTIE) (No. 20217910100010, Development of Monitoring and Diagnosis Technology based on Configuration Management for Buried Piping Damages of Operating Nuclear Power Plants).

**Conflicts of Interest:** The authors declare no conflict of interest.

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
