# Peer review of "Effect of Electrode Spacing on the Detection of Coating Defects in Buried Pipelines Using Direct Current Voltage Gradient Method"

_coatings, doi:10.3390/coatings13081471_

Round 1
Reviewer 1 Report
The manuscript is well-structured and the topic is researched in depth, which provides valuable insights into the effect of detection electrode spacing and detection voltage on DCVG detection signal magnitude.
(1) The results are presented in a structured manner with sufficient reference to the figures. However, there is room for providing more context. When presenting the findings, it would be helpful to briefly recap the research question or hypothesis that is being addressed by the results.
(2) The use of FEM simulation in your results is indeed insightful, however, it would be beneficial to delve deeper into how these simulations were conducted, and what specific parameters were taken into account. More transparency in this regard would greatly aid in the comprehension and reproducibility of your work.
(3) While the discussion section aptly interprets the findings, it could benefit from a broader perspective. Incorporating comparisons with prior research in the field would improve the manuscript's depth. For example, how do your results compare with previous studies on the same or related topics? Are there any differences or similarities, and how might they be explained?
(4) In the conclusion section, it would be helpful to emphasize the practical implications of your findings. While the conclusions are succinct and directly tied to your study, they could be made more compelling by suggesting potential applications of your research in real-world scenarios or specifying how this could guide future research directions.
(5) The conclusions drawn from the research are logical and directly derived from the results, but they lack a bit in future perspectives. Speculating on the broader implications, potential future work or next steps to take in light of the conclusions would give your research a forward-looking angle.
Author Response
We attached the answer.

Reviewer 2 Report
The article is interesting and very relevant. However, there are a few remarks that need to be corrected.
1. Introduction
- It is necessary to delete the last paragraph in the introduction. In its place, the purpose of the study should be clearly stated.
2. Materials and Methods
- There is a missing data about the experimental medium. Was it distilled water or soil? It is more likely that the authors used soil. What was the composition of the soil? I recommend adding this information to subsection 2.1.
3. Figures
- Figure 3 is a bit overlapping with the text. Please correct it.
- Figure 7 please make this figure readable. It may be necessary to increase the size of the figures.
4. Conclusions
- In my opinion, the first conclusion looks obvious. Could the authors please replace it?
5. Other comments
- I ask that the authors remove the merging of words in the following places: line 124: ±2m; line 154: 1.3m; line 183: (... 1.8m); (a) 1.0V, (b) 1.5V, (c) 3.0V, (d) 4.5V; line 185: Δ1m; line 187: Δ2m; line 189: Δ3m; line 214: 4.5V; line 229: 1.0V; lline 231: Δ3m; line 235: 1.0V; line 237: Δ1m; line 238: Δ2m and Δ3m; line 241: 4.5V, Δ1m; line 242: Δ2m and Δ3m; line 251: (a) 1.0V, (b) 1.5V, (c) 3.0V, (d) 4.5V; line 270: 1.0V; line 283: (... 1.8m); (a) 1.0V, (b) 1.5V, (c) 3.0V, (d) 4.5V.
- change equation (1) (line 122) on Equation (1) similar to the line 211 (Equation (1)).
- mostly in the text, the authors use the abbreviation of the word meters as m. It is requested that the authors make appropriate changes for the following lines: 1.8 meters (line 106); 1 to 1.5 meters (line 337).
Author Response
We attached the answer.

Reviewer 3 Report
This is the comments on the Manuscript to: Coatings (ISSN 2079-6412)
Manuscript ID: coatings-2553855
Type of manuscript: Article
Title: Effect of Electrode Spacing on the Detection of Coating Defects in Buried Pipelines using Direct Current Voltage Gradient Method
Author: Seung-Heon Choi , Young-Ran Yoo * , Young-Sik Kim *
Rate the Manuscript:
Significance to field and specialization of Coatings (ISSN 2079-6412)”
journal: good.
In the article it has been presented the results of a defect investigation of the buried piping is subject to soil corrosion and in which way can prevent by combining coatings and cathodic protection to maximize corrosion control.
It has been established even with both methods, coatings are subject to damage from external factors and various causes.
Buried piping may expose the metal and alter the current flow, which in turn causes corrosion.
The main conclusions:
The detection reliability increased as the noise signal eliminates at the optimum detection electrode spacing. However, the detection reliability decreased at higher selection electrode spacing as the noise signal and detected signals together eliminates. The location of the detected signal shifted as the spacing of the detection electrodes increased due to the change in the detection reference point and signal magnitude.
Scientific content: good.
Originality: good.
Clarity and presentation: acceptable.
Appropriateness for Journal: appropriate subject matter for the Coatings (ISSN 2079-6412)
Need for rapid publication: no.
1. What is the main question addressed by the research?
In this study analyzed the effect of detection electrode spacing on the direct current voltage gradient (DCVG) magnitude formed for coated pipelines buried in the soil. The DCVG was measured using a real-time coating defect detection system.
2. Do I consider the topic original or relevant in the field? Does it
address a specific gap in the field?
Yes.
3. What does it add to the subject area compared with other published material?
Yes.
4. What specific improvements should the authors consider regarding the methodology? What further controls should be considered?
The FEM model simulates to prove it, and then the result was compared to the measured DCVG magnitude. When the spacing of the detection electrodes increased, the detected signal and signal location changed.
5. Are the conclusions consistent with the evidence and arguments presented and do they address the main question posed?
Yes.
6. Are the references appropriate?
May be concider the next papers to compare the fracture mechanics approaches to the problem described: Workability Assessment of Structural Steels of Power Plant Units in Hydrogen Environments: Strength of Materials - 2009, vol. 41, - No 1. - P. 52-57. https://doi.org/10.1007/s11223-009-9097-4 ; Strength of welded joints of Cr-Mn steels with elevated content of nitrogen in hydrogen-containing media // Materials Science – 2009, N 1, p. 97-107. https://doi.org/10.1007/s11003-009-9166-7
The references are appropriate.This research based on 28 modern scientific works.
7. Please include any additional comments on the tables and figures.
No comments.

This is the comments on the Manuscript to: Coatings (ISSN 2079-6412)
Manuscript ID: coatings-2553855
Type of manuscript: Article
Title: Effect of Electrode Spacing on the Detection of Coating Defects in Buried Pipelines using Direct Current Voltage Gradient Method
Author: Seung-Heon Choi , Young-Ran Yoo * , Young-Sik Kim *
Rate the Manuscript:
Significance to field and specialization of Coatings (ISSN 2079-6412)”
journal: good.
In the article it has been presented the results of a defect investigation of the buried piping is subject to soil corrosion and in which way can prevent by combining coatings and cathodic protection to maximize corrosion control.
It has been established even with both methods, coatings are subject to damage from external factors and various causes.
Buried piping may expose the metal and alter the current flow, which in turn causes corrosion.
The main conclusions:
The detection reliability increased as the noise signal eliminates at the optimum detection electrode spacing. However, the detection reliability decreased at higher selection electrode spacing as the noise signal and detected signals together eliminates. The location of the detected signal shifted as the spacing of the detection electrodes increased due to the change in the detection reference point and signal magnitude.
Scientific content: good.
Originality: good.
Clarity and presentation: acceptable.
Appropriateness for Journal: appropriate subject matter for the Coatings (ISSN 2079-6412)
Need for rapid publication: no.
1. What is the main question addressed by the research?
In this study analyzed the effect of detection electrode spacing on the direct current voltage gradient (DCVG) magnitude formed for coated pipelines buried in the soil. The DCVG was measured using a real-time coating defect detection system.
2. Do I consider the topic original or relevant in the field? Does it
address a specific gap in the field?
Yes.
3. What does it add to the subject area compared with other published material?
Yes.
4. What specific improvements should the authors consider regarding the methodology? What further controls should be considered?
The FEM model simulates to prove it, and then the result was compared to the measured DCVG magnitude. When the spacing of the detection electrodes increased, the detected signal and signal location changed.
5. Are the conclusions consistent with the evidence and arguments presented and do they address the main question posed?
Yes.
6. Are the references appropriate?
May be concider the next papers to compare the fracture mechanics approaches to the problem described: Workability Assessment of Structural Steels of Power Plant Units in Hydrogen Environments: Strength of Materials - 2009, vol. 41, - No 1. - P. 52-57. https://doi.org/10.1007/s11223-009-9097-4 ; Strength of welded joints of Cr-Mn steels with elevated content of nitrogen in hydrogen-containing media // Materials Science – 2009, N 1, p. 97-107. https://doi.org/10.1007/s11003-009-9166-7
The references are appropriate.This research based on 28 modern scientific works.
7. Please include any additional comments on the tables and figures.
No comments.
Author Response
We attached the answer.
